# Tai Chi Training Evokes Significant Changes in Brain White Matter Network in Older Women

**DOI:** 10.3390/healthcare8010057

**Published:** 2020-03-09

**Authors:** Chunlin Yue, Liye Zou, Jian Mei, Damien Moore, Fabian Herold, Patrick Müller, Qian Yu, Yang Liu, Jingyuan Lin, Yuliu Tao, Paul Loprinzi, Zonghao Zhang

**Affiliations:** 1College of Physical Education and Sport Science, Soochow University, Suzhou 215021, China; y_chunlin002@163.com (C.Y.); JianMei_Zp@163.com (J.M.); taoyuliu@suda.edu.cn (Y.T.); 2Exercise and Mental Health Laboratory, Shenzhen University, Shenzhen 518060, China; liyezou123@gmail.com (L.Z.); yuqianmiss@163.com (Q.Y.); linjingyuan921@126.com (J.L.); 3Department of Health, Exercise Science and Recreation Management, The University of Mississippi, University, MS 38677, USA; dcmoore3@go.olemiss.edu (D.M.); pdloprin@olemiss.edu (P.L.); 4Research Group Neuroprotection, German Center for Neurodegenerative Diseases (DZNE), Leipziger Str. 44, 39120 Magdeburg, Germany; Fabian.herold@dzne.de (F.H.); Patrick.Mueller@dzne.de (P.M.); 5Department of Neurology, Medical Faculty, Otto von Guericke University, Leipziger Str. 44, 39120 Magdeburg, Germany; 6Department of Kinesiology and Program in Neuroscience, Indiana University, Bloomington, IN 9 47405, USA; YL82@indiana.edu

**Keywords:** exercise, Tai Chi, DTI, brain network of white matter, small world attributes

## Abstract

*Background:* Cognitive decline is age relevant and it can start as early as middle age. The decline becomes more obvious among older adults, which is highly associated with increased risk of developing dementia (e.g., Alzheimer’s disease). White matter damage was found to be related to cognitive decline through aging. The purpose of the current study was to compare the effects of Tai Chi (TC) versus walking on the brain white matter network among Chinese elderly women. *Methods:* A cross-sectional study was conducted where 42 healthy elderly women were included. Tai Chi practitioners (20 females, average age: 62.9 ± 2.38 years, education level 9.05 ± 1.8 years) and the matched walking participants (22 females, average age: 63.27 ± 3.58 years, educational level: 8.86 ± 2.74 years) underwent resting-state functional magnetic resonance imaging (rsfMRI) scans. Diffusion tensor imaging (DTI) and graph theory were employed to study the data, construct the white matter matrix, and compare the brain network attributes between the two groups. *Results:* Results from graph-based analyses showed that the small-world attributes were higher for the TC group than for the walking group (*p* < 0.05, Cohen’s d = 1.534). Some effects were significant (*p* < 0.001) with very large effect sizes. Meanwhile, the aggregation coefficient and local efficiency attributes were also higher for the TC group than for the walking group (*p* > 0.05). However, no significant difference was found between the two groups in node attributes and edge analysis. *Conclusion:* Regular TC training is more conducive to optimize the brain functioning and networking of the elderly. The results of the current study help to identify the mechanisms underlying the cognitive protective effects of TC.

## 1. Introduction

Aging is associated with serious changes in neurocognition. For instance, age-related cognitive decline can start as early as in one’s 20s and 30s and becomes more obvious among older adults [1]. When the age-related cognitive decline becomes sufficiently serious, it is highly associated with increased risk of developing dementia (e.g., Alzheimer’s disease) [2,3,4]. As cognition relies on proper brain functioning, it is natural to assume that the aging-related changes in cognition may be accompanied by age-related changes in the brain, too. Indeed, solid evidence shows that aging leads to considerable changes in functional connectivity and grey matter as well as white matter in the human brain [5,6]. In this context, a recent study showed that there is a significant ageing-related loss of white matter volume in fronto-striatal projections [7]. The white matter plays a key role in cognitive processes because it connects different grey matter regions throughout the human brain [8]. The important role of white matter in cognition is buttressed by the evidence that white matter damage was found to be related to cognitive decline through aging and that reduced integrity of white matter is associated with worsening of cognitive performance (e.g., executive functions, information processing speed) [9,10,11]. Furthermore, older individuals are more cognitively deficient than other individuals. Such differences were thought to be caused by brain networks with cognitively deficient older individuals tending to recruit a similar brain network as young adults but used it ineffectively, whereas cognitively deficient older adults compensated for cognitive decline by reorganizing their brain network [12]. Thus, it is of great importance to investigate whether aging influences brain white matter networks so we can determine methods to slow down these processes via lifestyle changes, such as physical exercise [8]. 

Notably, it was observed that (i) one year of aerobic exercise intervention positively influenced white matter structure and cognition [13]; (ii) a six-month dance intervention increased white matter volume in healthy older adults [14]; (iii) individuals who performed life-long physical exercises preserved their white matter integrity [15]; (iv) white matter microstructure mediated the relationship between cardiorespiratory fitness and cognitive performance (i.e., spatial memory) [16]. This evidence suggests that physical training and a relatively high level of cardiorespiratory fitness (e.g., achieved through regular physical training) are beneficial to preserving white matter integrity and cognition. However, no evidence determined the type of physical exercise (e.g., aerobic exercise, resistance exercise, motor-cognitive exercise) that is the most beneficial to preserve white matter integrity and cognition. The current study examines if dual-task training, which includes physical/or motor activity in combination with cognitive demands, is more effective in improving cognitive functions than a single task [17].

The literature provides evidence suggesting that motor-cognitive exercise (e.g., Tai-Chi, dancing) could be the most beneficial type of physical exercise to preserve or improve neurocognition in older individuals because they combine motor (physical) and cognitive demands [18,19]. Tai-Chi (TC) is a cognitive-motor exercise, which is commonly performed at mild-to-moderate exercise intensities [20,21]. Its choreographed routine typically consists of graceful, slow, fluid movements and it is performed in coordination with deep breathing, relaxation, and mental focus [22,23]. Such unique features have attracted people worldwide in pursuit of health and longevity, especially the frail elderly who experience functional decline [24,25]. Early studies have extensively investigated the health benefits of TC, suggesting that it effectively improved physical (e.g., balance, lower-limb strength, flexibility, and cardiovascular fitness) [26,27] and mental (e.g., stress, anxiety, pain, and depression) health outcomes of different age groups [28,29].

Scholars have also been interested in examining the neurocognitive benefits of TC [30]. Notably, the majority of early studies mainly focused on the effects of TC on behavioral outcome measures related to cognition [31,32]. With recent advances in neuroimaging techniques, scholars started to pay great attention to understanding the neurobiological mechanisms and processes underlying the cognitive protective effects of TC, suggesting that patients with brain damage demonstrate neuropsychological improvements after practicing TC [33]. Despite the brain decline through age, previous studies have indicated that the regional homogeneity of brain functions and cognition were significantly different between TC practitioners and TC-naïve individuals. Specifically, TC was shown to improve brain functions among older adults [34]. Furthermore, after a 12-week TC training, low frequency fluctuations of the brain increased in the dorsal prefrontal cortex in the TC group and higher prefrontal cortex activity was associated with better behavioral performance in the TC group [35,36]. However, little research has examined the changes in brain white matter due to TC practice, and compared it to a different form of regular physical exercise, such as walking, which is considered to be most feasible and low cost among many physical exercises. Hence, the current study aimed to examine whether TC practice and walking practice changes brain white matter and the brain network.

## 2. Materials and Methods

### 2.1. Participants

From 2017 to 2019, healthy elderly women were recruited to attend this study in Suzhou, China. Volunteers were screened against the exclusion criteria, including metal implants, abnormal hearing, serious physical diseases, family history of mental illnesses, drug or alcohol abuse, musculoskeletal diseases, injuries caused by sports, and claustrophobia. Forty-two females were included: (1) 20 practitioners (62.81 ± 3.02 years) in the TC group reporting exercise training exceeding 6 years, with 90 min × 5 sessions per week and education level of 9.05 ± 1.96 years; (2) 22 participants (63.55 ± 3.04 years) in Walking group where they regularly performed walking for 6 years, with 90 min × 5 sessions per week and educational level of 8.73 ± 2.21 years. Research assistants explained the purpose of the experimental procedures. All participants who volunteered to engage this study completed the written informed consent, which was approved by the ethics committee of the university (Approval No. ECSU-2019000209). Furthermore, all study procedures were in accordance with the latest version of the Declaration of Helsinki.

### 2.2. MRI Data Acquisition

The MRI images were captured by a Philips 3.0T MRI scanner with standard 32 channel head coil array in the MRI room of the Second Affiliated Hospital of Soochow University, Suzhou. During the whole scanning process, we used cushions to minimize head motion of participants and earplugs to reduce noise generated by the scanning. Then diffusion tensor imaging (DTI) data and T1-weighted terms data of all the participants were collected. Diffusion tensor imaging data were required by echo plane imaging sequences (EPI) with field of view: 220 × 220 mm^2^; matrix: 112 × 109 mm^2^; TR: 6000 ms; TE: 95ms and directions: 32. The number of layers was 50 and the slice thickness was 3 mm. Participants were scanned continuously using sagittal three dimensional T1-weighted high resolution magnetization-prepared rapid gradient echo sequence (MPRAGE) for their whole brains: 155 sagittal slices, voxel size: 0.625 × 0.625 × 1 mm^3^, TR: 7.1 ms, TE: 3.5 ms, flip angle: 8°, slice thickness: 1.0 mm; field of view: 220 × 220 mm^2^; matrix = 352 × 352 mm^2^.

### 2.3. Statistical Analysis

#### 2.3.1. Image Preprocessing

The data processing software PANDA (Positioning And Navigation Data Analyst) based on FSL (FMRIB Software Library) 5.0.9 was used for preprocessing and network construction (Figure 1). First, we converted DTI data from DICOM (Digital Imaging and Communications in Medicine) format to NIfTI (Neuroimaging Informatics Technology Initiative), and then corrected eddy current and head motion to eliminate the influence of gradient coil. Skull images were removed and the scalp and other non-brain tissue structures were stripped. The estimation of dispersion tensor model and the fractional anisotropy (FA) of every voxel were calculated. Fiber assignment by a continuous tracking (FACT) algorithm was used to reconstruct the direction of white matter fiber bundles in the brain network. High FA values indicate tight connections between the microstructures. Conversely, the lower FA value usually indicates white matter damage. All traces were calculated based on voxel seed point and the traced streamlines were terminated when the fold angle of the fibers was greater than 45 degrees or the FA value was less than 0.2.

#### 2.3.2. Construction of Deterministic Fiber Tracking Network

Nodes and edges are two basic elements of the brain network. In this study, the whole brain was divided into 90 regions of interest as 90 nodes by using an automated anatomical labeling (AAL) atlas, and each unilateral hemisphere contained 45 nodes. For the edge connection between two brain regions, one brain region has a fiber connection to the other and terminates in it, which is called a fibrous connection. The FA image was registered with the corresponding T1-weighted image in the original space by affine transformation. Then, the structural image was matched to the standard template through the nonlinear transformation. After the last two steps, the AAL atlas from the standard space could be mapped back to the individual space of each subject through inverse transformation. At the same time, we recorded the number of fiber connections between each pair of brain regions of every participant and stored the values to construct a 90 × 90 matrix.

#### 2.3.3. Characteristic Analysis of Topological Attributes of Complex Brain Networks 

We used the GRETNA (GRaph thEoreTical Network Analysis) 2.0 toolbox and the topological attributes of brain functional networks were calculated with the number of fibers with a minimum threshold of 10 to reduce the influence of false edges on network attributes. In the process of complex network research, regular networks and random networks have been used to describe and simulate the complex real system. The shortest path length and global efficiency reflect the capability to transmit global information about the network. Short paths indicate high global efficiency of the networks as well as the high efficiency information transmission between nodes in the network [37].

#### 2.3.4. Statistical Calculation

The statistical analysis was performed using SPSS (SPSS Inc., Chicago, IL, USA). We calculated a two-sample two-tail t-test (if the data was normally distributed) or Mann–Whitney (if the data was not normally distributed) in order to compare baseline demographic characteristics (see Table 1). To investigate the relationship between white matter brain network parameters and cognitive performance, we conducted a correlational analysis using Pearson’s correlation coefficient. The correlation coefficients were rated as follows: 0–0.19: no correlation; 0.2–0.39: low correlation, 0.40–0.59: moderate correlation; 0.60–0.79: moderately high correlation; ≥ 0.80: high correlation. The level of statistical significance was set in all statistical tests to α = 0.05.

## 3. Results

### 3.1. Demographic Data 

No significant differences were observed between the two groups in age, education level, handedness, MoCA, and MMSE scores. Detailed information is displayed in Table 1 and Table 2. Participants in walking group showed significantly lower scores in 2-back test (*p* = 0.016, Cohen’s *d* = 0.784), and significantly longer response times (*p* = 0.037, Cohen’s *d* = −0.664) in 2-back test compared with participants in Tai Chi group. 

### 3.2. fMRI Results

In both groups, λ (normalized characteristic path length of network) was close to one, and γ (normalized clustering coefficient) was greater than one, resulting in the small-world parameter greater than one (the white-matter correlation network of both groups followed a small-world property). There was no significant difference in the length of characteristic path between networks (*p* = 0.235), but the normalized clustering coefficient of the network was significantly higher in the TC group than in the Walking group (*p* < 0.05). Consequently, the small-world parameter (brain white matter network) was found to be greater in the TC group than in the walking group and the observed difference was greatly significant (*p* = 0.000). Changes in network parameters are depicted in Table 3 and Figure 2.

When looking into the relationship between small-world properties at the fMRI scan and individual performance, we found that the small-world attribute (Sigma) exhibited a low, negative correlation with reaction time (r (42) = −0.313; *p* = 0.044) in the 2-back condition and a moderately high, positive correlation with accuracy score (r (42) = 0.673; *p* ≤ 0.000) across all subjects from the two groups (Figure 3). The other computed correlations did not reach statistical significance and thus they were not reported.

## 4. Discussion

Previous experimental work has shown that TC can improve multiple forms of psychological well-being and influence the functional architecture of the brain in older adults [34,38]. However, so far only a few studies have examined the effects of TC on the human brain and available studies have focused on changes in functional brain activation during standardized cognitive tests or structural brain changes in response to a long-term TC training [36,39]. To the best of our knowledge, we are the first to investigate in a cross-sectional study the influence of long-term TC training on brain white matter and the brain network in comparison to walking training. Results of the current study indicate that TC can improve small-world attributes compared to walking (*M* = 5.02 ± 0.35 vs. 4.50 ± 0.33, *p* < 0.05, Cohen’s d = 1.534). The significant effects were huge. Although the aggregation coefficient and local efficiency attributes expressed positive trends in favor of TC compared to walking, these results were not statistically significant (*p* > 0.05).

The principal focus of the present paper was whether TC compared to walking could influence characteristics of brain white matter. Statistical evidence showed that TC could attenuate small-world attributes. Watts and Strogatz (1998), in their work have creatively and quantitatively described the small-world characteristics to define this kind of network with small, average short path and large clustering coefficients similar to the corresponding random network as in the small world network, which can be represented by two proportions [40]. The network with small world characteristics has high local efficiency and global efficiency, and the network can transfer information effectively both locally and globally. The results showed that TC affected Sigma in the small-world parameter. Moreover, Sigma might be linked to better individual cognitive performance in the 2-back task. The N-back test is considered to be an important paradigm for measuring the updating function in working memory [41]. Friedman et al. (2006) found that the updating function of working memory is most closely related to higher cognitive activities compared with other central executive functions. This finding aligns with similar results from several other studies on this general topic [42]. Wei et al. (2014) evaluated the effects of TC on intrinsic human brain architecture in older adults [34]. Age-matched older adults separated into two groups (TC versus healthy controls matched for sex, age, and education) completed a computerized flanker-type test measuring different behavioral aspects of attention. That is, participants had to rapidly and accurately respond to stimuli presented on a computer screen using either hand (left or right). The target stimulus was an arrow pointing in either direction (left or right) flanked by additional stimuli on each side. Furthermore, participants were instructed to press the left mouse button with their left thumb or the right mouse button with their right thumb as fast as possible when the target arrow pointed to the left or right, respectively. Significant decreases were detected by the fMRI in the left anterior cingulate cortices and the right superior frontal cortices of the dorsal lateral prefrontal cortices of the TC group compared to the controls. However, increases in the right post-central gyrases were observed in TC groups relative to the controls [34]. In this experiment, TC expressed greater functional brain activity within the post-central gyrases which may indicate a benefit of TC training. Moreover, TC training produced a greater beneficial effect on executive and non-executive cognitive functions compared to brisk walking, and this finding may be related to the inherent demands of TC (e.g., higher cognitive demands of TC compared to walking) [43].

Due to the mind-body component of TC, the combination of meditation with slow movements, deep rhythmic breathing, and relaxation has been shown to influence different regions of the human brain in a multitude of ways [44]. For example, TC is believed to influence the slow frequency fluctuations in resting brain activity by generating numerous combinations of oscillatory waves. As a result, changes in brain activity (throughout different regions) can be observed and quantified. Therefore, we speculate that the differences between TC and walking may be due to the different exercised-induced characteristics placed upon the body. As previously mentioned, TC involves a calming mind-body component that differs from walking (excitatory), thus producing different mechanisms acting upon the human brain and its associated networks.

In support of the present findings, it appears that TC may attenuate neural network changes in the human brain, therefore influencing the age-associated cognitive decline in elderly populations. Neural networks with high clustering and short path lengths are the direct benefit of long-term TC training, thus making the transmission of global and local information more efficient from network to network. From a public health perspective, our findings suggest that a positive influence of TC on white matter brain network could be of great interest because evidence suggests that white matter brain changes occur in aging and in neurological diseases (e.g., Alzheimer disease, Multiple sclerosis) [45,46]. For instance, alterations in brain white matter occurring in individuals suffering from Alzheimer disease and individuals with mild cognitive impairment (MCI) show white matter changes prior to development of dementia [9]. Hence, given our findings that long-term TC training positively influences white matter brain network and cognitive performance, it seems reasonable to speculate that physical interventions using TC training could be a promising strategy in dementia prevention. This idea is supported by the finding that engaging in long-term TC training significantly improved executive function performance in older adults with MCI [46,47]. With regard to Multiple sclerosis (MS), evidence in the literature indicates that (1) brain white matter changes are associated with changes in cognitive performance in MS [48,49] and (2) TC interventions in MS improve both motor-cognitive abilities (e.g., balance) and quality of life [26,50]. Whether such cognitive and motor-cognitive performance improvements after long-term TC training are caused by changes in brain white matter networks among other factors (e.g., upregulation of neurotrophic factors, grey matter changes) is a promising area for further investigations.

Our findings should be interpreted in light of the limitations of the current study and in light of the work from other laboratories, as few research studies have examined the effects of TC on the brain white matter network. A limitation of the current study is the homogenous sample of elderly women which inhibits the generalization of our findings. Likewise, considering the small sample size, until data is replicated using a larger sample size, results should be interpreted with caution. Future research should consider investigating the effects of TC on elderly men to determine gender influences in the long-term TC effects on changes in brain network activity. A strength of our investigation, however, is the comprehensive assessment of functional brain activity utilizing rsfMRI to capture changes in human brain network activity in response to long-term TC training.

## 5. Conclusions

In conclusion, our findings have shown that long-term TC training is more conducive to optimize the brain structure and to promote the efficient brain function network of older women. Long-term TC training produced significant changes in white matter small-world attributes. Changes as such may indicate improvements in the efficiency and transmission of neural data between human brain networks. This long-term TC training would prove useful in elderly populations where cognitive decline is prevalent. In contrast, non-significant changes were also noted for the aggregation coefficient, global efficiency and local efficiency attributes in favor of the TC group. Collectively, these attributes are used to measure the efficiency of neural transmission between networks in the human brain. Furthermore, findings may suggest that rsfMRI can be an appropriate tool for measuring the effects of TC on the characteristics of brain white matter in the human brain. As such, findings from this study encourage further investigations to test whether long-term TC training could be a promising strategy to protect against cognitive decline and alterations of white matter brain networks, which both occur in aging and in neurological diseases such as Multiple sclerosis, mild cognitive impairment, Alzheimer’s disease, and dementia.

## Figures and Tables

**Figure 1 healthcare-08-00057-f001:**
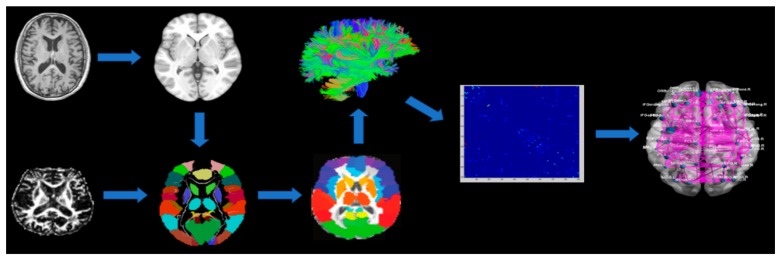
Construct flow chart of white matter network by using diffusion magnetic resonance imaging (MRI). Note: Individual fractional anisotropy (FA) image was registered to structural image, which were then non-linearly registered to ICBM152 standard space. Then, a transform matrix was obtained. The automated anatomical labeling (AAL) anatomical marker map was registered to individual spaces by means of the above two inverse transformation matrix steps. Whole brain white matter fibers were reconstructed as shown in (C). Each participant’s weighted network (G) was created by calculating the number of fibers which connected each pair of brain regions. The brain network matrix of two groups was constructed at the group level, and then the brain network attributes were calculated.

**Figure 2 healthcare-08-00057-f002:**
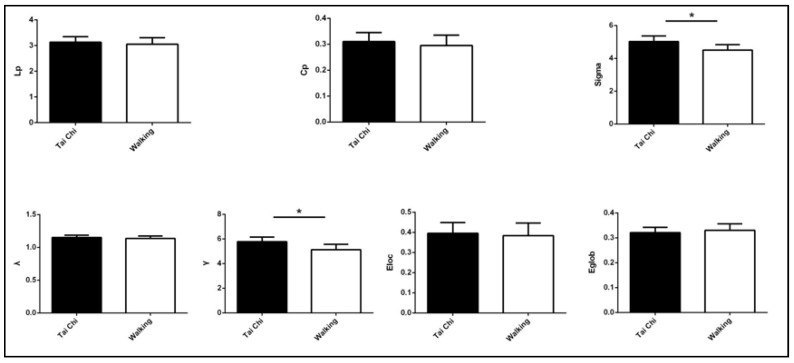
Comparison of brain networks between Tai Chi and Walking groups (**p* < 0.05).

**Figure 3 healthcare-08-00057-f003:**
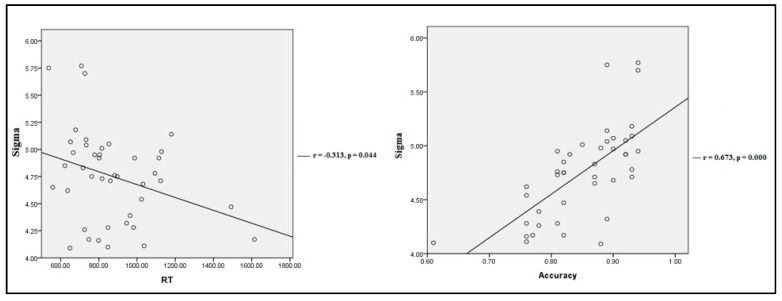
Partial correlation between Sigma at the fMRI scan and individual performance in 2-back test for all subjects controlling for age and years of education. Each circular dot represents the data from one participant.

**Table 1 healthcare-08-00057-t001:** Demographic Data.

	Tai Chi	Walking	*t*	*p*	*Cohen’s d*
	(*n* = 20)	(*n* = 22)			
Age, years	62.9 ± 2.4	63.27 ± 3.6	−0.393	0.193 ^a^	−0.121
Education, years	9.05 ± 1.8	8.86 ± 2.74	0.188	0.074 ^b^	0.082
Handedness (left/right)	0/20	0/22			
MMSE	28.5 ± 1.1	28.14 ± 1.0	1.1	0.636 ^b^	0.342
Moca	28.4 ± 1.5	27.5 ± 1.5	1.94	0.83 ^b^	0.6
RT(1-back)ms	629.26 ± 185.58	685.06 ± 111.94	0.133	0.196 ^a^	−0.364
RT(2-back)ms	761.03 ± 146.61	855.73 ± 138.39	2.198	0.037 ^a^	−0.664
Accuracy(1-back)	0.95 ± 0.03	0.93 ± 0.08	1.078	0.30 ^a^	0.331
Accuracy(2-back)	0.91 ± 0.04	0.87 ± 0.06	2.522	0.016 ^a^	0.784

Abbreviation: a—The *p* value was obtained using a two-sample two-tailed t-test; b—The *p*-value was obtained using non-parametric (Mann–Whitney) test. Note: MMSE: The Mini-Mental State Exam; MoCA: The Montreal Cognitive Assessment.

**Table 2 healthcare-08-00057-t002:** Topological parameters of brain functional networks used in this study.

Network Properties	Characters	Descriptions
Small-world properties	*C_p_*	The clustering coefficient of a network that is the average of the clustering coefficient, Cp-nodal, over all nodes. It measures the extent of local cluster or cliquishness of the network.
	*L_p_*	The characteristic path length of a network that is the average minimum number of connections linking any two nodes of the network. It measures the extent of overall routing efficiency of the network.
	*E_loc_*	The local efficiency of a network that is the average of the local efficiency, Eloc-nodal, over all nodes. It measures the mean local efficiency of the network.
	*E_glob_*	The global efficiency of a network that is the inverse of the harmonic mean of the minimum path length between any two nodes. It measures the extent of information propagation through the whole network.
	*S, K*	The sparsity or the cost to build a network.
Degree distribution	*a*	A scalar parameter, which reflects the extent that the node degree spans within a network.
	*k_c_*	A cutoff value, which evaluates the extent of an exponential decay.
Nodal properties	*k_nodal_*	The number of edges linking a single node.
	*C_p-nodal_*	The nodal clustering coefficient that measures the extent of inter-connectivity among the neighbors of the node.
	*E_loc-nodal_*	The nodal local efficiency that measures the extent of information transmission among the neighbors of the node.
	*E_nodal_*	The nodal global efficiency that measures the extent of information transmission of the node with all other nodes in the network.

(Notes: *C_p_*: clustering coefficient; *L_p_*: characteristic path length; *E_loc_*: local efficiency; *E_glob_*: global efficiency; *S, K*: sparsity (S) and wiring cost (K); *a*: a scalar parameter, a; *K_c_*: a cutoff degree, Kc; *K_nodal_*: degree of a node; *C_p-nodal_*: nodal cluster coefficient; *E_loc-nodal_*: nodal local efficiency; *E_nodal_*: nodal efficiency).

**Table 3 healthcare-08-00057-t003:** Comparison of network attributes between Tai Chi group and Walking group.

	Tai Chi Group		Walking Group		T	*p*	Cohen’s d
	M	SD	M	SD			
Lp	3.131	0.215	3.051	0.255	1.071	0.291	0.339
Cp	0.31	0.035	0.295	0.04	1.236	0.224	0.399
sigma	5.021	0.345	4.503	0.33	4.835	0.000 *	1.534
λ	1.151	0.036	1.137	0.037	1.207	0.235	0.384
γ	5.774	0.373	5.125	0.448	4.951	0.000 *	1.574
Eloc	0.395	0.054	0.384	0.062	0.609	0.546	0.189
Eglob	0.321	0.021	0.33	0.026	−1.233	0.225	−0.38

Notes: Lp: Shortest Path Length; Cp: Aggregation Coefficient; sigma: small-world Attribute; λ: normalized characteristic path length of network; γ: normalized clustering coefficient; Eloc: Local Efficiency. Eglob: Global Efficiency; M = mean; SD = standard deviation. **p* < 0.05.

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
