# Peer review of "Tai Chi Training Evokes Significant Changes in Brain White Matter Network in Older Women"

_healthcare, 2020, doi:10.3390/healthcare8010057_

Round 1

Reviewer 1 Report

I am reviewing the paper entitled “Tai Chi Training Evokes Significant Changes in Brain White Matter Network in Older Women” for Healthcare.  The authors did an excellent job on this paper.  I read this paper as my students were doing a peer review and I told them that this paper was the best one I have read in a while (and I review 22-26 papers per year) and I also told them that we had to get access to an fMRI because this paper showed that fMRIs can determine important biological effects of physical and meditational practices like Tai Chi.  As I thought the paper was excellent, I only have suggestions for improving English.  I will make those points from the beginning to the end of the paper.

Lines 21-22 should say “relevant and it can start as early as middle age.”  Line 24 should say “The purpose of the current study”.  Line 27 should say “Tai Chi practitioners”.  Line 32 should say “Results from graph-based analyses showed that the small-world attributes were higher for the TC group than for the walking group”.   Line 34 should say “attributes were also higher for the TC group than for the walking group”. Lines 37-38 should say “brain functioning and networking of the elderly. The results of the current study help to identify the mechanisms underlying the cognitive protective effects of TC.” 

Line 42-43 should say “For instance, age-related”.  Line 44 should say “When age-related …serious, it is”.  Line 46 should say “relies on proper brain functioning, it is”. Lines 47-48 should say “Indeed, solid evidence shows”.  Line 49 should say “a recent study showed”. Line 51 should say “matter plays a key”. Lines 55-59 should say “Furthermore, older individuals are more cognitively deficient than other individuals. Such differences were thought to be caused by brain networks with cognitively deficient older individuals tending to recruit a similar brain network as young adults but used it ineffectively, whereas cognitively deficient older adults compensated for cognitive decline …their brain networks”. Lines 60-61 should say “influences brain white matter networks, so we can determine methods to slow down these processes via lifestyle changes, such as”.

Line 62 should say “1 year of aerobic…influenced…”  Line 63 should say “intervention increased”.  Line 64 should say “individuals who performed”. Line 65 should say “preserved…microstructure mediated”. Line 67 should say “training and a relatively high level”. Line 68 should say “preserving white matter”. Line 69-70 should say “However, no evidence determined the type of…exercise that is the”.  Line 71 should say “The current study examines if”.  Line 72 should say “cognitive demands, is”.

Line 74 should say “The literature provides evidence suggesting”. Lines 78-79 should say “is a cognitive-motor exercise, which…exercise intensities”.  Line 80 says “movements and it is”. Line 82 should say “who experience”.  Line 83 should say “investigated the health benefits of”.  At this point, the authors need to provide the additional information necessary to describe the health benefits of what? Line 86 should say “examining the …benefits of TC”. Line 88 should say “advances in neuroimaging techniques”. Line 90 should say “damage demonstrating”. Line 92 should say “indicated that”. Line 93 should say “Specifically, TC”. Line 94 should say “after a 12-week TC”. Line 95 should say “TC group and higher…performance in the TC group”. Line 98 should say “compared it to…exercise, such as walking, which”.

Line 99 should say “study aimed”. Line 100 should say “whether TC practice”. Line 107 should say “the TC group reporting exercise training exceeding 6 years”. Line 110 should delete “Prior to the study” and it should start “Research”.  Line 110 should say “assistants explained the purpose of the experimental”. Line 112 should say “volunteered to engage in this study”. Line 116 should say “The MRI…by a Philips”. Line 120 should say “sequences”. Line 122 should say “and the slice thickness was”. Line 131 should say “First,”. Line 132 should say “Skull images”. Line 134 should say “Fiber assignment by a continuous”. Line 135 should say “algorithm was used”. Line 136 should say “High FA values indicate tight connections between the microstructures.”

Line 142 should say “FA images were registered to structural images, which were then”. Line 143 should say “Then, a transform…obtained. The AAL”. Line 144 should say “individual spaces…two inverse…matrix steps.” Line 145 should say “matter fibers were”. Line 152 should say “hemisphere contained 45 nodes. For the edge”. Line 162 should say “networks”. Line 166 should say “global-information”. Lines 166-168 should say “Short paths indicate high global efficiency of the networks as well as the high efficiency information transmission between nodes in the network”. Line 171 should say “if the data were not normally”. Line 174 should say “a correlational analysis…Coefficients.”

Line 180 should say “differences were”. Line 182 should place a space after “Table 1”. Line 182 should say “group showed…scores in the 1-back”. Line 183 should say “tests…response times” and delete the comma between “test” and “compared”. Line 187 should say “two-tailed…The p-value”.

Line 196 should say “A significant difference was found in”. Line 197-198 should say “network was significantly higher in the TC group than in the Walking group”.  Line 199 should say “TC group than in the Walking group”. Line 200 talks about greatly significant, but no such concept exists in statistics. Rather, a result is either significant or not.

Line 222 should say “first to investigate…study the…influence on”. Line 223 should say “comparison to walking.” Line 224 should delete the word “respectively”. Line 225 should say “Although the aggression”. Line 226-227 should say “Walking, these”. Line 229-230 should say “small-world” something. The authors need to finish the point. Line 230 should put a year in parentheses after the authors. Line 231-232 should say “characteristics to define…small average short paths and large clustering coefficients”. Line 234 should say “characteristics have”. Line 235 should say “The results”. Lines 236-237 should say “linked to better individual cognitive performance in the 2-back task. The N-back-test” . Line 238 should put the year after the author and “et al.”. The same is true on line 240. Line 253-254 should say “Moreover, TC training produced”. Line 255 should say “walking, and this finding may be related”. Lines 261-262 should say “can be observed and quantified. Therefore, we speculate that”. Line 271 should say “suggest a positive”. Line 272 should say “could be of great interest because evidence suggests that”. Lines 274-276 should say “For instance, alterations in brain white matter occur…disease and individuals…impairment (MCI) show white matter changes prior”.

Lines 281-283 should say “(MS), evidence in the literature indicates that 1) brain whiter matter changes…and 2) TC interventions in MS improve both”. The clause on line 285 starting “among other” and ending “changes)” should be moved to line 286 after “networks”. Line 290 should say “women, which inhibits the generalization of our findings”. Line 292 should say “elderly man to determine gender influences in the long-term TC effects on”. Line 201 says “This would”. This what? Line 306 should delete the “the” at the end of the line. Line 307 should say “whether long-term”. Line 308 should put a comma after “networks” and line 309 should put a comma after “diseases”.  

Author Response

Dear reviewer:

Thank you for your suggestions from which we have benefited immensely. We have revised this manuscript according to your suggestions and we believed that the article have become more logical and fluent. We have marked the revised parts as red in the article.

  1. Lines 21-22 should say “relevant and it can start as early as middle age.” 

Thanks for your suggestion and it has been revised.

  1. Line 24 should say “The purpose of the current study”. 

       Thanks for your suggestion and it has been revised.

  1. Line 27 should say “Tai Chi practitioners”. 

       Thanks for your suggestion and it has been revised.

  1. Line 32 should say “Results from graph-based analyses showed that the small-world attributes were higher for the TC group than for the walking group”.   

       Thanks for your suggestion and it has been revised.

  1. Line 34 should say “attributes were also higher for the TC group than for the walking group”.

       Thanks for your suggestion and it has been revised.

  1. Lines 37-38 should say “brain functioning and networking of the elderly. The results of the current study help to identify the mechanisms underlying the cognitive protective effects of TC.” 

       Thanks for your suggestion and it has been revised.

  1. Line 42-43 should say “For instance, age-related”. 

       Thanks for your suggestion and it has been revised.

  1. Line 44 should say “When age-related …serious, it is”. 

       Thanks for your suggestion and it has been revised.

  1. Line 46 should say “relies on proper brain functioning, it is”.

       Thanks for your suggestion and it has been revised.

  1. Lines 47-48 should say “Indeed, solid evidence shows”. 

       Thanks for your suggestion and it has been revised.

  1. Line 49 should say “a recent study showed”.

       Thanks for your suggestion and it has been revised.

  1. Line 51 should say “matter plays a key”.

       Thanks for your suggestion and it has been revised.

  1. Lines 55-59 should say “Furthermore, older individuals are more cognitively deficient than other individuals. Such differences were thought to be caused by brain networks with cognitively deficient older individuals tending to recruit a similar brain network as young adults but used it ineffectively, whereas cognitively deficient older adults compensated for cognitive decline …their brain networks”.

       Thanks for your suggestion and it has been revised.

  1. Lines 60-61 should say “influences brain white matter networks, so we can determine methods to slow down these processes via lifestyle changes, such as”.

       Thanks for your suggestion and it has been revised.

  1. Line 62 should say “1 year of aerobic…influenced…” 

       Thanks for your suggestion and it has been revised.

  1. Line 63 should say “intervention increased”. 

       Thanks for your suggestion and it has been revised.

  1. Line 64 should say “individuals who performed”.

       Thanks for your suggestion and it has been revised.

  1. Line 65 should say “preserved…microstructure mediated”.

       Thanks for your suggestion and it has been revised.

  1. Line 67 should say “training and a relatively high level”.

       Thanks for your suggestion and it has been revised.

  1. Line 68 should say “preserving white matter”.

       Thanks for your suggestion and it has been revised.

  1. Line 69-70 should say “However, no evidence determined the type of…exercise that is the”. 

       Thanks for your suggestion and it has been revised.

  1. Line 71 should say “The current study examines if”. 

       Thanks for your suggestion and it has been revised.

  1. Line 72 should say “cognitive demands, is”.

       Thanks for your suggestion and it has been revised.

  1. Line 74 should say “The literature provides evidence suggesting”.

       Thanks for your suggestion and it has been revised.

  1. Lines 78-79 should say “is a cognitive-motor exercise, which…exercise intensities”.  

       Thanks for your suggestion and it has been revised.

  1. Line 80 says “movements and it is”.

       Thanks for your suggestion and it has been revised.

  1. Line 82 should say “who experience”. 

       Thanks for your suggestion and it has been revised.

  1. Line 83 should say “investigated the health benefits of”.  At this point, the authors need to provide the additional information necessary to describe the health benefits of what?

       Thanks for your suggestion and it has been revised as the following.

     Early studies have extensively investigated the health benefits of TC, suggesting that it effectively improved physical (e.g., balance, lower-limb strength, flexibility, and cardiovascular fitness) and mental (e.g., stress, anxiety, pain, and depression) health outcomes of different age groups.

  1. Line 86 should say “examining the …benefits of TC”.

       Thanks for your suggestion and it has been revised.

  1. Line 88 should say “advances in neuroimaging techniques”.

       Thanks for your suggestion and it has been revised.

  1. Line 90 should say “damage demonstrating”.

       Thanks for your suggestion and it has been revised.

  1. Line 92 should say “indicated that”.

       Thanks for your suggestion and it has been revised.

  1. Line 93 should say “Specifically, TC”.

       Thanks for your suggestion and it has been revised.

  1. Line 94 should say “after a 12-week TC”.

       Thanks for your suggestion and it has been revised.

  1. Line 95 should say “TC group and higher…performance in the TC group”.

       Thanks for your suggestion and it has been revised.

  1. Line 98 should say “compared it to…exercise, such as walking, which”.

Thanks for your suggestion and it has been revised.

  1. Line 99 should say “study aimed”.

     Thanks for your suggestion and it has been revised.

  1. Line 100 should say “whether TC practice”.

     Thanks for your suggestion and it has been revised.

  1. Line 107 should say “the TC group reporting exercise training exceeding 6 years”.

     Thanks for your suggestion and it has been revised.

  1. Line 110 should delete “Prior to the study” and it should start “Research”. 

     Thanks for your suggestion and it has been revised.

  1. Line 110 should say “assistants explained the purpose of the experimental”.

     Thanks for your suggestion and it has been revised.

  1. Line 112 should say “volunteered to engage in this study”.

     Thanks for your suggestion and it has been revised.

  1. Line 116 should say “The MRI…by a Philips”.

     Thanks for your suggestion and it has been revised.

  1. Line 120 should say “sequences”.

     Thanks for your suggestion and it has been revised.

  1. Line 122 should say “and the slice thickness was”.

     Thanks for your suggestion and it has been revised.

  1. Line 131 should say “First,”.

     Thanks for your suggestion and it has been revised.

  1. Line 132 should say “Skull images”.

     Thanks for your suggestion and it has been revised.

  1. Line 134 should say “Fiber assignment by a continuous”.

     Thanks for your suggestion and it has been revised.

  1. Line 135 should say “algorithm was used”.

     Thanks for your suggestion and it has been revised.

  1. Line 136 should say “High FA values indicate tight connections between the microstructures.”

     Thanks for your suggestion and it has been revised.

  1. Line 142 should say “FA images were registered to structural images, which were then”.

     Thanks for your suggestion and it has been revised.

  1. Line 143 should say “Then, a transform…obtained. The AAL”.

     Thanks for your suggestion and it has been revised.

  1. Line 144 should say “individual spaces…two inverse…matrix steps.”

     Thanks for your suggestion and it has been revised.

  1. Line 145 should say “matter fibers were”.

     Thanks for your suggestion and it has been revised.

  1. Line 152 should say “hemisphere contained 45 nodes. For the edge”.

     Thanks for your suggestion and it has been revised.

  1. Line 162 should say “networks”.

     Thanks for your suggestion and it has been revised.

  1. Line 166 should say “global-information”.

     Thanks for your suggestion and it has been revised.

  1. Lines 166-168 should say “Short paths indicate high global efficiency of the networks as well as the high efficiency information transmission between nodes in the network”.

     Thanks for your suggestion and it has been revised.

  1. Line 171 should say “if the data were not normally”.

     Thanks for your suggestion and it has been revised.

  1. Line 174 should say “a correlational analysis…Coefficients.”

     Thanks for your suggestion and it has been revised.

  1. Line 180 should say “differences were”.

     Thanks for your suggestion and it has been revised.

  1. Line 182 should place a space after “Table 1”.

     Thanks for your suggestion and it has been revised.

  1. Line 182 should say “group showed…scores in the 1-back”.

     Thanks for your suggestion and it has been revised.

  1. Line 183 should say “tests…response times” and delete the comma between “test” and “compared”.

     Thanks for your suggestion and it has been revised.

  1. Line 187 should say “two-tailed…The p-value”.

     Thanks for your suggestion and it has been revised.

  1. Line 196 should say “A significant difference was found in”.

     Thanks for your suggestion and it has been revised.

  1. Line 197-198 should say “network was significantly higher in the TC group than in the Walking group”.  

     Thanks for your suggestion and it has been revised.

  1. Line 199 should say “TC group than in the Walking group”.

     Thanks for your suggestion and it has been revised.

  1. Line 200 talks about greatly significant, but no such concept exists in statistics. Rather, a result is either significant or not.

       Thanks for your suggestion and the according p-value (p=0.000), which has been marked as red, could be found in line 5 of Table 3.

  1. Line 222 should say “first to investigate…study the…influence on”.

     Thanks for your suggestion and it has been revised.

  1. Line 223 should say “comparison to walking.”

     Thanks for your suggestion and it has been revised.

  1. Line 224 should delete the word “respectively”.

     Thanks for your suggestion and it has been deleted.

  1. Line 225 should say “Although the aggression”.

     Thanks for your suggestion and it has been revised.

  1. Line 226-227 should say “Walking, these”.

     Thanks for your suggestion and it has been revised.

  1. Line 229-230 should say “small-world” something. The authors need to finish the point.

     Thanks for your suggestion and It should be small-world Attribute.

  1. Line 230 should put a year in parentheses after the authors.

     Thanks for your suggestion and it has been revised.

  1. Line 231-232 should say “characteristics to define…small average short paths and large clustering coefficients”.

     Thanks for your suggestion and it has been revised.

  1. Line 234 should say “characteristics have”.

     Thanks for your suggestion and it has been revised.

  1. Line 235 should say “The results”.

     Thanks for your suggestion and it has been revised.

  1. Lines 236-237 should say “linked to better individual cognitive performance in the 2-back task. The N-back-test” .

     Thanks for your suggestion and it has been revised.

  1. Line 238 should put the year after the author and “et al.”. The same is true on line 240.

     Thanks for your suggestion and it has been revised.

  1. Line 253-254 should say “Moreover, TC training produced”.

     Thanks for your suggestion and it has been revised.

  1. Line 255 should say “walking, and this finding may be related”.

     Thanks for your suggestion and it has been revised.

  1. Lines 261-262 should say “can be observed and quantified. Therefore, we speculate that”.

     Thanks for your suggestion and it has been revised.

  1. Line 271 should say “suggest a positive”.

     Thanks for your suggestion and it has been revised.

  1. Line 272 should say “could be of great interest because evidence suggests that”.

     Thanks for your suggestion and it has been revised.

  1. Lines 274-276 should say “For instance, alterations in brain white matter occur…disease and individuals…impairment (MCI) show white matter changes prior”.

     Thanks for your suggestion and it has been revised.

  1. Lines 281-283 should say “(MS), evidence in the literature indicates that 1) brain whiter matter changes…and 2) TC interventions in MS improve both”. The clause on line 285 starting “among other” and ending “changes)” should be moved to line 286 after “networks”.

     Thanks for your suggestion and it has been revised.

  1. Line 290 should say “women, which inhibits the generalization of our findings”.

     Thanks for your suggestion and it has been revised.

  1. Line 292 should say “elderly man to determine gender influences in the long-term TC effects on”.

     Thanks for your suggestion and it has been revised.

  1. Line 301 says “This would”. This what?

     The long-term TC training.

  1. Line 306 should delete the “the” at the end of the line.

     Thanks for your suggestion and it has been revised.

  1. Line 307 should say “whether long-term”.

     Thanks for your suggestion and it has been revised.

  1. Line 308 should put a comma after “networks”.

     Thanks for your suggestion and it has been revised.

  1. Line 309 should put a comma after “diseases”.  

     Thanks for your suggestion and it has been revised.

Reviewer 2 Report

The authors compared brain white matter networks in a sample of elderly women practicing either Tai Chi or walking. The topic is of interest, the introduction is well referenced and the methods appear to be sound. The paper is well written and results are adequately discussed. Limitations of the study are correctly acknowledged. 

I only have minor observations:

  • The numbers referring to the titles of the sections are wrong (all are 1) and should be amended.
  • At page 2, line 61: reference for this sentence "..due to lifestyle changes such as physical exercise" is not formatted in the same way as the others and the sentence ends with a comma.
  • At page 3 line 131: is "NIFIT" the correct format or did the authors mean "NIfTI"?

Author Response

Dear reviewer:

Thank you for your suggestions from which we have benefited immensely. We have revised this manuscript according to your suggestions and we believed that the article have become more logical and fluent. We have marked the revised parts as green in the article.

  1. The numbers referring to the titles of the sections are wrong (all are 1) and should be amended.

Thanks for your suggestion and it has been revised.

  1. At page 2, line 61: reference for this sentence "..due to lifestyle changes such as physical exercise" is not formatted in the same way as the others and the sentence ends with a comma.

Thanks for your suggestion and it has been revised. (The same comment has    been given by reviewer 1.)

  1. At page 3 line 131: is "NIFIT" the correct format or did the authors mean "NIfTI"?

Thanks for your suggestion and it means NifTI.

Reviewer 3 Report

In table 2, the effect sizes, in order, are 0.338, 0.398, 1.54, 0.383, 1.57, 0.189, and 0.379.  The significant effects are huge (Cohen, 1992 places .80 or larger as large for an effect size) and the others are in the small to medium range (0.20 to 0.50) where an effect size of 0.50 could be "seen" by a careful observer. 

In Figure 3, one wonders if there is heterogeneity of variance as a function of the independent variable, this should be tested.

In Table 2, all of the abbreviations used should be explained in the footnote to the Table since Tables are supposed to be stand alone.

In Table 2, when I ran a t-test for Accuracy (1-back) I got t = 1.05 and a non-significant p value (.299).  When I ran a t-test for Accuracy (2-back) I got t = 2.515 rather than 2.7 and my p was .016, not .009.  This may be a graphpad vs. SPSS method of calculation but the results should be re-checked by the authors.

The discussion section should include discussion of the effect sizes, which should also be calculated for Table 1 (I did them but will leave them for the authors to figure out).

The discussion in the abstract should include mention of the relative effect sizes. 

On page 2, line 64/65 the (iii) is used twice; I think the second time it should be (iv) rather than (iii).

On page 3, line 103, China should be added to the address of the city since many readers may not be familiar with Suzhou.

On page 3, line 137 I'd suggest "indicates" or "suggests" rather than "intimates"

Page 3, line 132 it was not clear what was meant by "Skull were removed and.... were stripped."  Does this refer to photo editing?

Author Response

Dear reviewer:

Thank you for your suggestions from which we have benefited immensely. We have revised this manuscript according to your suggestions and we believed that the article have become more logical and fluent. We have marked the revised parts as blue in the article.

  1. In table 2, the effect sizes, in order, are 0.338, 0.398, 1.54, 0.383, 1.57, 0.189, and 0.379.  The significant effects are huge (Cohen, 1992 places .80 or larger as large for an effect size) and the others are in the small to medium range (0.20 to 0.50) where an effect size of 0.50 could be "seen" by a careful observer. 

 Thanks for your comment and “Cohen's d” has been added to Table 3.

  1. In Figure 3, one wonders if there is heterogeneity of variance as a function of the independent variable, this should be tested.

Thanks for your comment and partial correlation analysis has been performed   with age and years of education as covariates.

  1. In Table 2, all of the abbreviations used should be explained in the footnote to the Table since Tables are supposed to be stand alone.

Thanks for your comment and all of the abbreviations used have been explained.

( Cp: clustering coefficient; Lp: characteristic path length; Eloc: local efficiency; Eglob: global efficiency; S,K: sparsity (S) and wiring cost (K); a: a scalar parameter, a; Kc: a cutoff degree, Kc; Knodal: degree of a node; Cp-nodal: nodal cluster coefficient; Eloc-nodal: nodal local efficiency; Enodal: nodal efficiency.)

  1. In Table 2, when I ran a t-test for Accuracy (1-back) I got t = 1.05 and a non-significant p value (.299).  When I ran a t-test for Accuracy (2-back) I got t = 2.515 rather than 2.7 and my p was .016, not .009.  This may be a graphpad vs. SPSS method of calculation but the results should be re-checked by the authors.

Thanks for your comment. It has been re-checked and “Cohen's d” has been added to Table 3.

  1. The discussion section should include discussion of the effect sizes, which should also be calculated for Table 1 (I did them but will leave them for the authors to figure out).

Thanks for your comment. “Cohen's d” has been added to Table 1. The discussion section already includes a discussion of the effect size.

  1. The discussion in the abstract should include mention of the relative effect sizes. 

Thanks for your comment. The abstract section already includes a discussion of the effect size.

  1. On page 2, line 64/65 the (iii) is used twice; I think the second time it should be (iv) rather than (iii).

Thanks for your comment and it has been changed to (iv).

  1. On page 3, line 103, China should be added to the address of the city since many readers may not be familiar with Suzhou.

Thanks for your comment and “China” has been added.

  1. On page 3, line 137 I'd suggest "indicates" or "suggests" rather than "intimates"

Thanks for your comment and we use “indicates” this time.

  1. Page 3, line 132 it was not clear what was meant by "Skull were removed and.... were stripped."  Does this refer to photo editing?

Thanks for your comment and it has been refined as “ Skull images were removed and the scalp and other non-brain tissue structure were stripped”.

Round 2

Reviewer 3 Report

1.  Abstract.  The term agglomeration is used here but elsewhere in the paper the term used is aggregation.  One way or the other, the same term should be used if the intent of the meaning is the same.

2.  Abstract.  The sentence "The significant effects were huge" is a misnomer.  It would be more accurate to say that "Some effects were significant (p < .001) with very large effect sizes."

3.  Abstract.  The abstract says that global efficiency was higher for the TC group but in Table 2 it appears to be lower.  This inconsistency needs to be resolved.

4.  Table 1 does not appear to have Cohen's d shown even though the authors' reply seemed to say it did.

5.  An explanation for the lambda and gamma labels in Table 3 should also be explained in the notes below that table. 

6.  Discussion; here it still seems that global efficiency is seen as higher for the TC group when it wasn't higher.

The paper is much improved but these details should still be corrected before acceptance.

Author Response

Dear reviewer:

Thank you for your suggestions from which we have benefited immensely. We have revised this manuscript according to your suggestions and we believed that the article have become more logical and fluent. We have marked the revised parts as blue in the article.

  1. The term agglomeration is used here but elsewhere in the paper the term used is aggregation.  One way or the other, the same term should be used if the intent of the meaning is the same.

Thanks for your comment and it has been changed to aggregation.

  1. The sentence "The significant effects were huge" is a misnomer.  It would be more accurate to say that "Some effects were significant (p < .001) with very large effect sizes."

Thanks for your comment and it has been changed to “ Some effects were significant (p < .001) with very large effect sizes.”

  1. The abstract says that global efficiency was higher for the TC group but in Table 2 it appears to be lower.  This inconsistency needs to be resolved.

Thanks for your kindly reminder and the according description about global efficiency in abstract has been removed.

  1. Table 1 does not appear to have Cohen's d shown even though the authors' reply seemed to say it did.

Thanks for your kindly reminder. Just want to emphasize that Table 1 only includes demographic information, which is unnecessary to add Cohen’s d. Instead, we have added the value of Cohen’s d in Table 3 where it shows comparison of network attributes between Tai Chi group and walking group.

  1. An explanation for the lambda and gamma labels in Table 3 should also be explained in the notes below that table. 

Thanks for your comment. The explanation for the lambda and gamma labels have been added ( λ: normalized characteristic path length of network; γ: normalized clustering coefficient ).

  1. Discussion; here it still seems that global efficiency is seen as higher for the TC group when it wasn't higher.

Thanks for your comment and it has been revised.
